# Validation of Methods for Assessment of Dust Levels in Layer Barns

**DOI:** 10.3390/ani13050783

**Published:** 2023-02-21

**Authors:** Solène Mousqué, Frédérique Mocz, Anja B. Riber

**Affiliations:** 1AgroSup Dijon, Higher National School of Agriculture and Food Science, 21079 Dijon, France; 2Epidemiology, Health and Welfare Unit, French Agency for Food, Environmental and Occupational Health & Safety (ANSES), 22440 Ploufragan, France; 3Department of Animal and Veterinary Sciences, Aarhus University, DK-8830 Tjele, Denmark

**Keywords:** dust assessment, dust sheet test, laying hens, validation

## Abstract

**Simple Summary:**

Currently, veterinarians conducting animal welfare inspections lack validated methods for measuring or assessing dust levels in poultry barns. In the present study, we examined the validity of six methods for dust assessment in layer barns. The methods were either developed for the purpose or as a refinement of existing methods. Of the six methods examined, the dust sheet test with a test duration of 2–3 h was found to be the most promising method, showing a high validity. Furthermore, results indicated that if more steps are added to the scoring scale, further reduction of the test duration of the dust sheet test may potentially be possible without losing the validity, making the test more feasible for veterinary inspections. More research is needed to examine this hypothesis as well as the reliability that was not addressed in the present study.

**Abstract:**

The dust level is included in the animal welfare legislation of the European Union, implying assessment of dust levels during veterinary welfare inspections. This study aimed to develop a valid and feasible method for measuring dust levels in poultry barns. Dust levels were assessed in 11 layer barns using six methods: light scattering measurement, the dust sheet test with durations of 1 h and 2–3 h, respectively, visibility assessment, deposition assessment, and a tape test. As a reference, gravimetric measurements were obtained – a method known to be accurate but unsuitable for veterinary inspection. The dust sheet test 2–3 h showed the highest correlation with the reference method with the data points scattered closely around the regression line and the slope being highly significant (*p* = 0.00003). In addition, the dust sheet test 2–3 h had the highest adjusted R^2^ (0.9192) and the lowest RMSE (0.3553), indicating a high capability of predicting the true concentration value of dust in layer barns. Thus, the dust sheet test with a test duration of 2–3 h is a valid method for assessing dust levels. A major challenge is the test duration as 2–3 h is longer than most veterinary inspections. Nevertheless, results showed that potentially, with some modifications to the scoring scale, the dust sheet test may be reduced to 1 h without losing validity.

## 1. Introduction

Three terms are used to refer to what is commonly known as dust: aerosols, particulate matter (PM), and dust [1]. The term aerosol refers to “airborne particles, in either solid or liquid state, that are usually stable in a gas for at least a few seconds”. The term aerosol is synonymous with PM, whereas the term dust more specifically refers to ‘solid particles formed by mechanical disintegration of a material’.

The Council Directive 98/58/EC of 20 July 1998 concerning the protection of animals kept for farming purposes states that “dust levels […] must be kept within limits which are not harmful to the animals” [2]. Thus, in order to comply with animal welfare legislation, it is necessary to be able to measure dust levels in poultry houses. However, there is currently a gap of knowledge in regard to valid methods for measuring dust levels during poultry welfare inspections. Recently, at a workshop involving the competent authorities of the majority of the EU Member States, the participants were asked which requirements in the laying hen and broiler welfare legislation they thought were lacking a validated assessment method. The legal requirement on dust level came out in first place for both the laying hens and broilers [3]. Follow-up interviews with official veterinarian inspectors confirmed that the methods currently available are considered insufficiently validated and/or insufficiently standardised to be reliable [3]. In this respect, validity is the extent to which an indicator and its method of assessment is meaningful in terms of providing accurate information on a legal requirement concerning an animal or a group of animals, and reliability is the extent to which results are consistently the same between two or more observers or when the same observer repeats assessments after receiving reasonable training [4,5].

Currently, the method used most frequently by inspectors is a sensorial assessment where the dust level is assessed based on visibility in the barn and the sensation of dust in the inspector’s respiratory system [3]. It may involve (1) a visual assessment of the dust level in the air of the poultry house, (2) an inspectors’ sensation of irritation (assumed to be caused by dust) in their own eyes, throat, and lungs, and (3) signs of irritation of the birds’ eyes, throat and lungs, assumed to be caused by dust. However, the method is considered to be of limited validity and reliability by the inspectors themselves as it relies on high levels of subjectivity [3]. Most inspectors are also aware of the dust sheet test: a method described in the Welfare Quality Protocols for laying hens and broilers [6,7] and tested in the present study. Thus, the test can be carried out in a standardised way, but the validity of the test has not previously been examined. Furthermore, the test duration is 3 h, which is considered a challenge as most veterinary inspections are of shorter duration.

Dust levels may also be measured using dust measuring devices that mainly have two measuring principles: gravimetry and light scattering. When using a gravimetric device, the air to be assessed is drawn through an impactor or cyclone, and particles with a chosen diameter are collected on filters, which are weighed before and after loading to determine the mass of particles sampled [8]. Gravimetry is the gold standard for dust measurement, and devices exploiting this principle are therefore typically used as reference methods [9]. However, sampling dust by use of gravimetric devices is time-consuming, involving procedures both before and after the sampling. In contrast, light scattering devices determine the concentration of PM by directing the air stream to be sampled into a light scattering chamber. A beam of light is then directed onto the particles in the air, and a sensor measures the amount of light scattered by the particles [8]. The device uses a relationship between the scattered light and the mass concentration of the particles, which is usually pre-set at the factory using a standard type of dust [9]. Light scattering devices are portable, easy to use, and provide real-time data at a relatively low cost.

The aim of the present study was to investigate the validity of existing and new methods of measuring dust levels, which are feasible for use during veterinary inspections of animal welfare. As a reference method, the dust level was collected by the use of a gravimetric measuring device.

## 2. Materials and Methods

The study took place from May 2021 to August 2021 and involved 11 different layer barns located all over Denmark. The farms were selected from those suggested by the Danish egg companies and with the agreement of the farmers.

### 2.1. Animals

Barns were visited when the birds were 22 (n = 4), 23 (n = 1), 32 (n = 2), 55 (n = 3), and 74 (n = 1) weeks old. All barns had a multitier housing system with two levels of slatted platforms as well as perches available for the hens. The floor was fully covered with litter, consisting of either wood pellets, straw pellets, or wood shavings. The number of hens per barn varied from 20,000 to 30,000, divided into four to five sections of 4000 to 7500 hens by wire mesh (Figure 1). Both ends of the barns contained corridors that were divided from the adjacent sections by wire mesh. The number of entrances from the corridors into the sections was 3 or 4 at each end of the barn. The stocking density was nine birds/m^2^ of usable floor area. The barns had forced ventilation systems and provided no access to a veranda or outdoor area.

### 2.2. Methodology

In each barn, the dust level was assessed using six methods (explained in detail in the following sections): light scattering measurement, dust sheet test with a duration of 1 h, dust sheet test with a duration of 2–3 h, tape test, visibility assessment, and deposition assessment. These were compared to a reference method, i.e., gravimetric measurement. One observer collected all the data during the barn visits. The spots in the barns, where the different types of assessments took place, are indicated in Figure 1.

#### 2.2.1. Gravimetric Measurement—The Reference Method

As the reference method or gold standard, a gravimetric device was chosen: the Atmospheric Dust Captor device (*Capteur Atmosphérique de Poussière*, CAP 10; Arelco, Fontenay-Sous-Bois, France). Before each measurement, an unloaded filter was placed in a proofer at 40 °C for 12 h and then desiccated for 30 min. The filter was then weighed two times using a precise balance (A&D HR-120, Japan; readability: 0.0001 g). The average value was calculated as the filter weight. The device was cleaned and disinfected before each barn visit. The device was switched on as soon as entering the barn and placed on a tripod right at approximately 90 cm above floor level (placement: see Figure 1). The working period of the device was noted. This was determined by the duration of the barn visit and ranged between 2 and 3 h.

Following sampling, a similar drying procedure as the one prior to the visit was performed before the loaded filter was weighed. The weight difference between the loaded and unloaded filter equaled the amount of collected dust. The dust concentration was calculated as the mass of collected dust divided by the operating time of the device. The total dust concentration was presented as milligrams per cubic metre of air. This concentration was considered the “true” dust concentration in the barn.

#### 2.2.2. Light Scattering Measurement

The concentration of PM_10_ was measured with a device using the light scattering principle (Dylos Corporation, DC1700-PM PM_2.5_ / PM_10_, Riverside, CA, USA). The result was given in milligrams per cubic metre of air, i.e., similar to the gravimetric measurement. Measurements were obtained approximately 50 cm above the floor and were carried out once in front of each entrance (Figure 1). The observer waited for a minute after arrival at the measuring spots before switching on the device to allow dust whirled up by the arrival of the observer to settle. The PM_10_ concentration was displayed 10 s after switching on the device.

The device was cleaned and disinfected before each visit. However, the inside of the device cannot be fully cleaned, meaning that air passing through the inside of the device may potentially be contaminated by particles from previous farm visits. Therefore, it was chosen to tape a 100 L airtight plastic bag to the bottom opening of the device before each visit to avoid bringing in particles from previous farm visits. Once the measurements were completed, the device was placed in another airtight bag as once the device, and hence the fan inside is switched off, the air stored in the plastic bag taped to the bottom may escape through the top inlet.

#### 2.2.3. Dust Sheet Tests

The dust sheet tests were conducted as described in the Welfare Quality Protocol for laying hens [6]. Two black A5-size papers were placed horizontally on the closed feed chains just outside each entrance (i.e., out of reach of the birds; Figure 1). The papers were placed immediately after entering the barn. One paper for each spot was assessed and removed 1 h later (‘dust sheet test 1 h’), whereas the second paper was left for 2 to 3 h (‘dust sheet test 2–3 h’), depending on the duration of the farm visit. Finger strokes on the paper were used to assess the amount of dust accumulated on the paper while comparing it with clean paper. The scoring scale [6]:Score 0 = No or minimal evidence of dust (sheet has same colour as clean sheet);Score 1 = Isolated specks or a thin layer of dust on sheet is detectable (without comparing with a clean sheet, the test sheet still appears black but there is a slight colour difference between the two sheets);Score 2 = Dust covers the sheet; even without comparing with a clean sheet, it is clear that the test sheet is no longer black (i.e., there is a clear difference in colour between clean and test sheets).

#### 2.2.4. Tape Test

To overcome the challenge of being back at a particular spot at an exact time point (as in the dust sheet test with 1 h duration), a tape test was developed using the principle of the dust sheet test. The upper and most horizontal part of the right shoulder of the assessor was covered with black duct tape (approx. 10 cm × 5 cm). The tape was placed when entering the barn, and the level of dust deposited on it was assessed 1 h after entering the barn. The cloth of the coverall was gently pulled away from the neck in order to be able to see the tape without having to pull it off the coverall. The scoring scale used for the dust sheet test was applied, and the tape on the shoulder was compared with a clean piece of tape.

#### 2.2.5. Visibility Assessment

The level of dust in the air was assessed in the visibility test where the observer stood approximately 20 m from the end wall within each area between the multi-tier systems, facing the wall (Figure 1). The scoring scale used was developed based on the information gained during interviews with veterinary inspectors describing how they performed assessments of dust levels in poultry barns:Score 0: No or limited dust: No visible dust specks in the air;Score 1: Moderate dust level: Isolated dust specks are detectable, but visibility has only slightly or not notably decreased – details of the wall are easily recognisable;Score 2: High dust level: Many dust specks are detectable, and visibility has notably decreased – details of the wall are not easily recognisable.

#### 2.2.6. Deposition Assessment

The level of dust was assessed based on the amount of dust accumulated on four structures in the barn. The structures selected were not easily accessible to the birds but still in a position where dust could accumulate, e.g., the pipe above the drinker line. As far as possible, it was attempted to use similar structures in all barns. The test was performed near the spots used for the visibility assessments (Figure 1). The scoring scale used to assess the level of dust deposited on each structure was as follows:Score 0 = No or minimal evidence of dust (structure has same colour as a spot nearby that has been wiped clean);Score 1 = Isolated specks or a thin layer of dust (≤1 mm) is detectable (when comparing with a cleaned part of the structure; a slight colour difference appears between the two parts of the structure);Score 2 = Dust covers the structure (clear difference in colour between clean and test parts). Dust layer is >1 mm but <5 mm;Score 3 = As in score 2, but the dust layer is ≥5 mm.

### 2.3. Statistical Analysis

Data were analysed using R Studio Software [10], and a significance level of 0.05 was used for all statistical tests. Gravimetric measurements (i.e., the reference method) were obtained from only nine of the 11 farm visits due to maloperation of aparatus. In addition, due to restrictions posed as a consequence of COVID-19 and some unexpected events during the visits, some data from methods examined were not obtained. Therefore, the number of pairs between the reference method and each method was: light scattering measurement (n = 7), dust sheet test 1 h (n = 9), dust sheet test 2–3 h (n = 9), tape test (n = 9), visibility assessment (n = 8) and deposition assessment (n = 8).

Prior to analysis, the data from the reference method were log-transformed to obtain data normality. For the methods where the data were collected multiple times per barn visit, an average was calculated (i.e., all, except the tape test). A linear regression analysis was conducted separately for each method tested to assess whether the degree to which the dust level obtained with the different methods was in agreement with the concentration given by the reference method. The dust concentration measured by the reference method was used as the dependent variable, whereas the dust level obtained using the different methods under examination was used as the independent variable, i.e., we tested whether the actual dust level, given by the reference method, could be predicted from the one given by the method examined:*Log (reference method)* = *µ + α * [dust concentration from test i]*(1)
where α is the slope, *µ* is the intercept, and *i* one of the six methods examined.

Before analysing the results of the linear regressions, the three assumptions of a linear model were checked. The assumption of normality of the residuals was checked using the Shapiro test and visually by the QQ-plot. The hypothesis of the homogeneity of variances was checked using the Breusch-Pagan test and visually by plotting the residuals against the model’s predicted values for the observed values of the predictive variable (i.e., “residuals vs. fitted plot”). Finally, the hypothesis of non-correlation of the residuals was verified using the Durbin– Watson test.

To determine the significance of the correlation between the reference method and each of the methods under examination, it was chosen to focus on the *p*-value of the slope of the regression line instead of the slope of the regression line between the dust concentration measured by the reference method and the methods under examination. The reason for this was that a slope of the regression line equal to 1 was not representative of identical concentrations between the reference method and the methods under examination because the latter express dust concentrations in different units (except for the light scattering measurements).

A significant *p*-value (*p* < 0.05) of the slope of the regression line meant that the slope was different from 0, i.e., the values obtained by the reference method and the method under examination were associated. The adjusted coefficient of determination (adjusted R^2^) was used to describe the correlation of the measured dust levels between the reference method and the method under examination. The closer the adjusted R^2^ is to 1, the better the model is in terms of predicting the true concentration value. The Root-Mean-Square Error (RMSE) of the models was used to provide information about the performance of a model by allowing a term-by-term comparison of the actual difference between the estimated and the measured value. The lower the value of RMSE, the better the performance of the model.

## 3. Results

The dust concentration measured using the gravimetric principle varied between 0.3–20.3 mg/m^3^ in the layer barns visited (Figure 2).

The regression lines for all the methods under examination are shown in Figure 3A-F, and the *p*-values for the slopes of the regression lines, adjusted R^2^ and the RMSE are listed in Table 1.

The dust sheet test 2–3 h showed the highest correlation with the reference method as the data points were scattered closely around the regression line (Figure 3C), and the slope was highly significant (Table 1). In addition, the adjusted R^2^ for the dust sheet test 2–3 h also had the highest value of the methods examined (Table 1), being close to one, further indicating a strong correlation between the dust level given by the dust sheet test 2–3 h and the reference method. Concerning the RMSE, the dust sheet test 2–3 h had the lowest value (Table 1), again indicating that the dust sheet test 2–3 h was the method providing the best prediction of the true value of dust concentration in the layer barns.

In contrast, the slope of the regression line for the deposition assessment was not significant (Figure 3F; Table 1), i.e., the dust level given by this method was not correlated to that of the reference method. In line with this, the slope was negative, i.e., the higher the concentration measured using the reference method, the lower the level of dust accumulated. In addition, the method received the lowest and thus the poorest adjusted coefficient of determination.

Although the slopes of the regression lines were significant for the tape test (Figure 3D) and the visibility assessment (Figure 3E), the distributions of the data points around the regression lines clearly showed that these methods performed poorly at predicting the dust level.

The light scattering device (Figure 3A) and the dust sheet test 1 h (Figure 3B) performed intermediate in regard to the distributions of the data points around the regression lines. Both methods had a significant slope, although at a lower significance level as for the dust sheet test 2–3 h. However, the adjusted R^2^ and the RMSE were relatively low (only for dust sheet 1 h) and high, respectively, i.e., showing a weaker correlation between dust levels collected using these methods and the reference method than those collected using the dust sheet test 2–3 h. The plot of the residuals against the dust concentration values predicted by the model for the dust sheet test 1 h showed increased variance with increasing concentration of dust (Figure 4). This indicates that the validity of the dust sheet test 1 h may be improved if the scale is further developed into a more detailed protocol with additional categories, e.g., five levels instead of the current three levels.

## 4. Discussion

Methods for measuring dust levels in poultry barns are vital for several reasons. Firstly, without knowledge of the dust levels in poultry barns, initiatives focusing on reducing harmful levels of dust are at risk of not being prioritised. Dust from animal houses is an indoor pollutant with negative impacts on animal performance, health, and welfare [11,12,13], as well as on the health and welfare of animal caretakers [14]. In addition, dust is released outside animal houses by ventilation systems and is thus an outdoor pollutant. Poultry houses are, compared to other livestock houses, among those with the highest dust concentrations [15,16,17]. Intensive poultry production releases considerable amounts of dust to the atmosphere, accounting for around 55% of dust emissions from agriculture in Europe [18]. Dust contributes to the spread of diseases to other poultry houses in the vicinity and even to neighbouring human populations and is therefore a public health concern [11,19,20,21]. Secondly, legal requirements need to be measurable for the veterinary inspectors to be able to assess compliance with the legislation. In regard to the current legal requirement on dust, an open norm is used, i.e., no exact threshold value is given. However, since it states that the dust level must be kept within limits, which are not harmful to the animals [2], knowledge is needed both on threshold values for when the dust becomes harmful and on methods applicable for the assessment of dust levels. In addition, the methods applied need to be valid, reliable, and feasible for animal welfare assessments to be fruitful [22].

Six methods, considered relatively feasible for dust assessment during veterinary inspections, were examined in the present study for their validity. Although the number of farm visits was reduced compared to the original plan due to the COVID-19 health crisis and an avian influenza outbreak, the study revealed that the methods differed in potential. The most promising method was the dust sheet test, which is simple to perform and requires limited training, preparation, and investment as the only thing needed is pieces of black paper that can be discarded after each inspection. Based on the scattering of the data points closely around the regression line and good performance of all the statistical measures applied, i.e., significant p-value of the slope, high adjusted R^2,^ and low RMSE, the dust sheet test with a test duration of 2–3 h was shown to be a valid method of assessing dust levels in poultry barns. The validity of the dust sheet test has not previously been examined even though the test is included in the Welfare Quality^®^ assessment protocols for both laying hens and broilers [6,7], and therefore is applied in multiple studies of animal welfare (e.g., [23,24,25]). However, the long test duration is a challenge for its use during an official veterinary inspection, which typically only lasts approximately 1 h [3]. Therefore, continued development of the protocol for the dust sheet test is needed with the aim of reducing the duration without losing the validity of the test. Although an association between the dust sheet test 1 h and the reference method was found (i.e., significant *p*-value of the slope), the relatively low adjusted R^2^ and high RMSE showed that a test duration of 1 h is insufficient time to obtain results as satisfactory as those obtained after 2–3 h, but the increased variation with increasing dust concentration indicates that more steps on the scoring scale may be a solution to improve the validity of the method when shorter test durations are applied. More research is needed to examine this hypothesis. Furthermore, in the present study, we discuss the feasibility and examined the validity of the methods, whereas no data on reliability were obtained. The dust sheet test is prone to subjectivity as it is based on the observers’ visual assessment. Future studies should address the reliability of the dust sheet test within and among observers.

Light scattering measurements appeared initially to be a feasible solution for the assessment of dust levels in poultry barns. The method is widely used for scientific dust studies and air quality control (e.g., [26,27]), but not yet for veterinary animal welfare inspections. However, the specific device used in the present study turned out to be unsuitable for use in poultry barns due to the lack of possibility of sufficiently cleaning the inner chamber, posing a biosecurity risk. We solved this with the attachment of a plastic bag, but this was an impractical solution that limited the number of measurements that could be obtained. In terms of validity, the results were intermediate. Some of the measures supported validity, i.e., the significant p-value of the slope and low adjusted R^2^. However, from the regression line of the log-transformed dust concentration from the reference method as a function of the dust level obtained by light scattering measurements, it appears that the data obtained using this method did not follow a linear pattern but perhaps more a hyperbolic curve. To test this hypothesis, and hence further examine the validation of the method, more data points, i.e., a larger sample size, are needed. With the continued developments in technology, the idea of light scattering devices for measuring dust levels in poultry barns should be further pursued.

The other methods examined, i.e., the tape test, visibility assessment, and deposition assessment cannot be recommended. For the tape test and visibility assessment, it is clear from the regression analyses that the linear models were not well-adapted, including poor distribution of data points around the regression line, low adjusted R^2^ (tape test), and high RMSE (visibility test). However, due to the low number of data points, it was not possible to add additional parameters for a better model fit to be obtained. In any case, it is questionable whether extra data points will add more information, as the very low variability in the scores given when using these tests poorly reflects the variation of dust level concentration measured by the reference method in the barns. The idea of the tape test was to develop a method that allows the time of exposure to be consistent without the need for the observer or veterinary inspector to return to the specific spots where the dust sheets were placed at a specific time. The poor performance may have been due to the movements of the observer obstructing dust accumulation as compared to what is seen on stationary dust sheets. Regarding the visibility assessment, the protocol was an attempt to standardise the sensorial assessment currently performed by most veterinary inspectors [3]. Obviously, the test is very subjective as it highly depends on the vision of the observer and the observer’s interpretation of the sight. The idea of the deposition test also originated from interviews with veterinarian inspectors, where some informed us that they occasionally use the dust deposition on barn structures as an indicator of the dust level [3]. None of the measures obtained for the deposition test indicated the validity of the test, i.e., negative slope, non-significant *p*-value of the slope, low R^2,^ and high RMSE, and clearly, this method presents many challenges. Firstly, the barns are usually completely cleaned before new flocks are placed, so depending on the age of the birds at the time of the inspection there is more or less dust accumulated, which does not necessarily indicate the current airborne dust level. Secondly, it is difficult to standardise the spots for the assessment between barns as the housing systems and additional structures differ greatly between barns. Thirdly, it is difficult to find spots that are suitable for the assessment, as spots should be untouched by the birds and not influenced by a draft from the ventilation or air puffs from the birds, while at the same time allowing unhindered accumulation of dust.

## 5. Conclusions

Currently, veterinarians conducting animal welfare inspections lack validated methods for measuring or assessing dust levels in poultry barns. In the present study, we developed new methods or refined existing methods for dust assessment in layer barns. Of the six methods, the dust sheet test with a duration of 2–3 h was found to be the most promising method, showing a high validity. However, a test duration of 2–3 his longer than most veterinarian inspections. A solution may be modifications of the scoring scale as the results indicated that with more steps on the scoring scale, it may potentially be possible to further reduce the duration of the dust sheet test without losing the validity. More research is needed to examine this hypothesis as well as the reliability that was not addressed in the present study.

## Figures and Tables

**Figure 1 animals-13-00783-f001:**
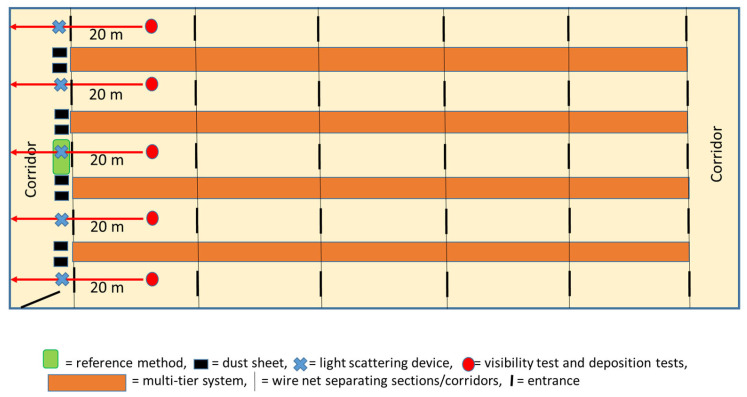
Spots of assessments or measurements in a common barn design.

**Figure 2 animals-13-00783-f002:**
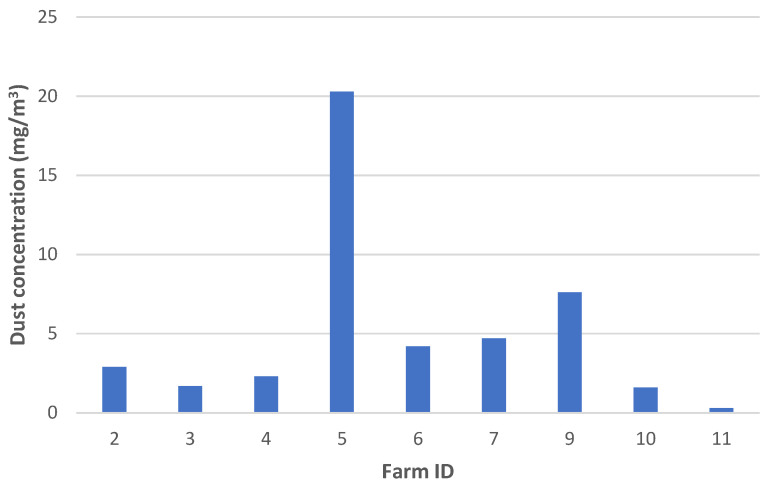
The total dust concentration (mg/m^3^) measured during the farm visits using the gravimetric principle (i.e., the reference method). No data were obtained during visits to Farms 1 and 8 due to failure of the device.

**Figure 3 animals-13-00783-f003:**
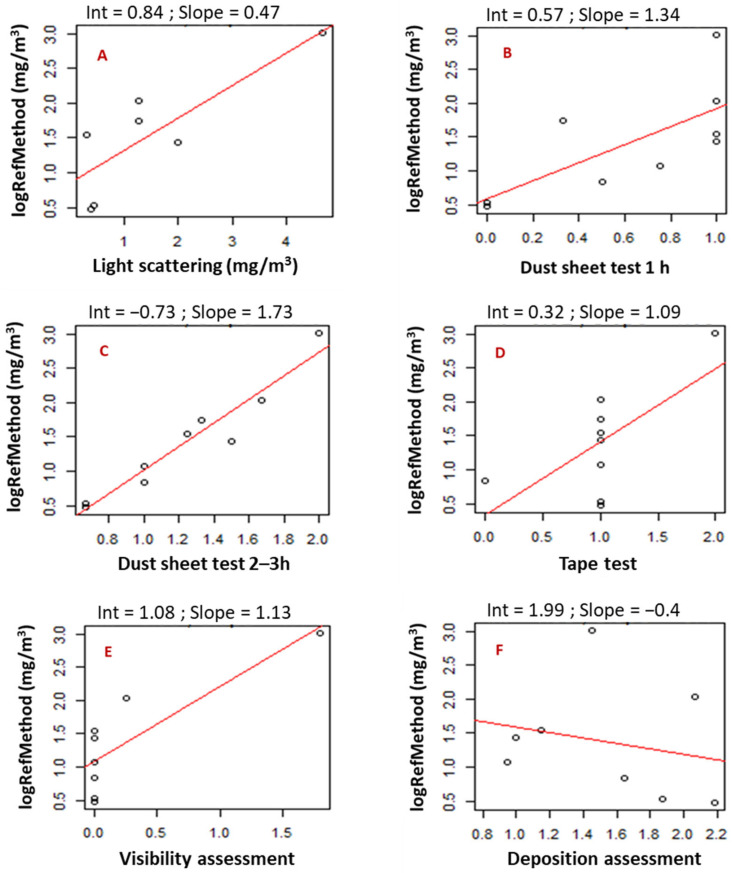
The regression lines (in red) for the log-transformed dust concentration from the reference method as a function of the dust level obtained using the methods under examination: (**A**) light scattering measurement, (**B**) dust sheet test 1 h, (**C**) dust sheet test 2–3 h, (**D**) tape test, (**E**) visibility assessment and (**F**) deposition assessment. The intercept (Int) and regression coefficient (slope) of each linear regression are given above the corresponding graph.

**Figure 4 animals-13-00783-f004:**
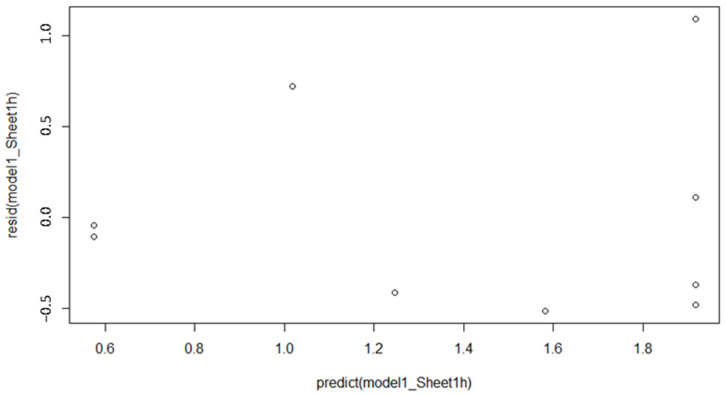
Plot of the residuals against the dust concentration values predicted by the model for the dust sheet test 1 h.

**Table 1 animals-13-00783-t001:** Results on the significance of the slopes of the regression lines for the log-transformed dust concentration of the reference method as a function of the dust concentrations of the methods under examination. The adjusted coefficient of determination (adjusted R^2^) and the Root-Mean-Square Error (RMSE) of each model are also given.

Method Examined	*p*-Value of Slope ^a^	Significance ^b^	Adjusted R^2^	RMSE
Light scattering measurements	0.02174	*	0.6206	0.7567
Dust sheet test 1 h	0.03118	*	0.4376	0.7987
Dust sheet test 2–3 h	0.00003	***	0.9192	0.3553
Tape test	0.04541	*	0.3801	0.4088
Visibility assessment	0.00994	**	0.6462	1.1112
Deposition assessment	0.58756	n.s.	−0.1062	0.6546

^a^ Each slope was tested against the nullhypothesis (i.e., being different from 0). ^b^
*p*-values of the slopes indicated as n.s. were not significantly different from 0, whereas star symbols indicate rejection of the null hypothesis at a significance level of 0.05 (*), 0.01 (**), or 0.001 (***).

## Data Availability

Data is contained in the article.

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
