# Peer review of "Validation of Methods for Assessment of Dust Levels in Layer Barns"

_animals, 2023, doi:10.3390/ani13050783_

Round 1

Reviewer 1 Report

Apologies Anja but I don't think this manuscript is suitable for this journal. Well aware of your work on poultry welfare, but this paper provides no data on "Animals". I suggest it is better suited to an air quality or environmental science journal or something like that.

Author Response

We thank the reviewer for taking time to review the paper. However, we completely disagree with Reviewer 1 regarding the comment that this "paper provides no data on "Animals"". Animal welfare assessments are based on the assessment of a range of welfare indicators. These may be animal-based (which we think is what reviewer 1 has in mind), resource-based (which is what we use in the current paper) or management based. Resource-based indicators are variables that are not measured in the animals but in their environment. In addition, the dust level may also indicate management issues, such as insufficient/improper settings of ventilation in the barn. 

Reviewer 2 Report

The study compared several methods to determine dust levels in poultry barns. The results are of high importance for animal welfare assessment, but also for public health. A valid and standardized method is required to provide recommendations for legal obligations, and investigate impact on animal health and determine cut-off values.

The background, methods, analysis and results are described clearly and in detail, the discussion is relevant. I recommend acceptance for publication, after addressing some minor suggestions:

Intro (L 53 ff) please provide definitions of validity and reliability

L 63: could you refer to the methods when describe the dust sheet test (or mention that this method was tested in your study)

L 126/143 Why is the paper for the dust sheet test placed immediately, while for the PM10 test the measurements commence after 1 minute? Would this have an influence on the outcome?

L 162: please clarify that you refer to the right shoulder of the coverall here (At first I thought you meant the right corner of the test paper). Please also clarify which kind of tape, plastic or paper or duct tape?

L 308: could you add “In regards to the *current* legal requirement on dust, an open norm is used..”

L321 Add a “be” before discarded

Discussion: How high is the impact of your sample size (which is rather low)? Could you add a thought on this aspect in the discussion? Would, with larger sample size, one of the methods become more/less reliable?

Author Response

We thank the reviewer for the positive review and the valuable comments. Below you find our response to the comments.

Intro (L 53 ff) please provide definitions of validity and reliability

RE: Definitions are now added.

L 63: could you refer to the methods when describe the dust sheet test (or mention that this method was tested in your study)

RE: Done.

L 126/143 Why is the paper for the dust sheet test placed immediately, while for the PM10 test the measurements commence after 1 minute? Would this have an influence on the outcome?

RE:  The papers were placed immediately in order to have the maximum possible exposure time for the 2-3 h Dust sheet test. Also, because they are at a level where they are not sensitive/subject to the movement of dust due to the observer's movement and therefore there was no need to wait. In contrast, the test with the light scattering device was performed closer to the ground, where the movement of dust due to our movement is the highest, and this was the reason for waiting 1 min before commencement of the light scattering test.

L 162: please clarify that you refer to the right shoulder of the coverall here (At first I thought you meant the right corner of the test paper). Please also clarify which kind of tape, plastic or paper or duct tape?

RE: Clarification added “the right shoulder of the assessor” and the type of tape, i.e. black duct tape.

L 308: could you add “In regards to the *current* legal requirement on dust, an open norm is used..”

RE: Done as suggested.

L321 Add a “be” before discarded

RE: Done.

Discussion: How high is the impact of your sample size (which is rather low)? Could you add a thought on this aspect in the discussion? Would, with larger sample size, one of the methods become more/less reliable?

RE: We agree that the sample size is rather low, and as explained in the manuscript, it was intended to be higher, but due to COVID and Avian Influenza we had to cancel some planned barn visits. We have added some speculations on how the sample size may have impacted the outcome of the light scattering, the visibility test and the tape test.

Reviewer 3 Report

The research increases the knowledge of dust measuring methods in poultry farms. The text is fluently written and fits well with the scope of the journal. I have only one comment for the authors to clarify before I can recommend this manuscript for publication: authors propose a set of statistical measures in the materials and methods (R2, MSPE), they do report them in the tables, but they don’t discuss all of them in the discussion section.

Author Response

We thank the reviewer for the positive review and the valuable comment. We have now elaborated the discussion to incorporate the point raised by the reviewer.

Round 2

Reviewer 1 Report

Sorry if I missed it, but how is this manuscript advancing our undertsanding of animals?